# CATS: COST-AUGMENTED TREE SEARCH FOR LLM-ASSISTED PLANNING

## ABSTRACT

While large language models excel at open-ended reasoning, they often struggle with cost-sensitive planning, either treating all actions as having equal cost or failing to stay within strict budgets. In this paper, we introduce Cost-Augmented Tree Search (CATS), a novel search approach that brings explicit cost-awareness into LLM-guided planning. Tight cost constraints push the planner to quickly identify infeasible solutions, while looser constraints encourage optimization for minimal cost. We benchmark top LLMs such as GPT-4.1 and Claude-Opus-4.1 against our CATS planner to evaluate their performance on a cost-augmented variant of BlocksWorld, where each action is assigned a specific budget and tasks must be completed under an overall budget constraint. Our experiments show that raw LLMs, such as Claude-Opus-4.1, often falter under tight budgets, whereas CATS consistently delivers strong performance with higher task success rates and better budget utilization. CATS provides an effective solution for budget-aware planning by combining the reasoning power of LLMs with structured search.

## 1 INTRODUCTION

Planning is a cornerstone of real-world decision-making, yet most research on LLM-assisted planning overlooks a critical factor: cost. Whether it is travel planning with budget constraints, scheduling meetings with transition times, or strategic resource allocation, cost considerations are paramount (Xie et al., 2024; Zheng et al., 2024; Kambhampati et al., 2024; Huang et al., 2024; Li et al., 2024; Wei et al., 2025). Some constraints are hard, e.g., missing a train departure, while others are soft, e.g., slightly exceeding a budget, but both demand cost-aware solutions. Existing LLM-based planners often assume all actions have the same cost, leading to suboptimal solutions. Our work bridges this gap by introducing cost-augmented planning, where models explicitly optimize for both task success and cost efficiency.

LLMs have demonstrated impressive reasoning capabilities, particularly in code generation, mathematical problem-solving, and logical reasoning (Shao et al., 2024; Ke et al., 2025; Hao et al., 2025). These breakthroughs suggest that LLMs could, in principle, handle constrained planning tasks by leveraging their emergent world understanding and step-by-step reasoning. However, while LLMs excel in open-ended reasoning, their ability to adhere to strict cost constraints remains understudied (Kambhampati et al., 2024). Can LLMs natively solve cost-sensitive planning problems, or do they require algorithmic enhancements to resolve constraints effectively? Our work investigates this question, testing both raw LLM performance and hybrid approaches that combine LLMs with structured search.

To evaluate on the cost setting, we augment the widely used BlocksWorld benchmark (Valmeekam et al., 2022) with explicit action costs to better study budget-aware planning. Each action is assigned a numerical cost, transforming the originally cost-agnostic benchmark into one that encourages cost-efficient solutions. This cost-augmented setting prioritizes low-cost valid plans, aligning more closely with real-world decision-making scenarios. An illustrative example of our cost-aware Budget-BlocksWorld is in Figure 1. We classify the difficulty of the task into three budget constraints, tight, loose, and unlimited. Tight constraints force planners to quickly recognize impossible solutions, while loose and unlimited encourage optimization for minimal cost. Building on this cost-augmented benchmark, we next turn to search-based planning methods.

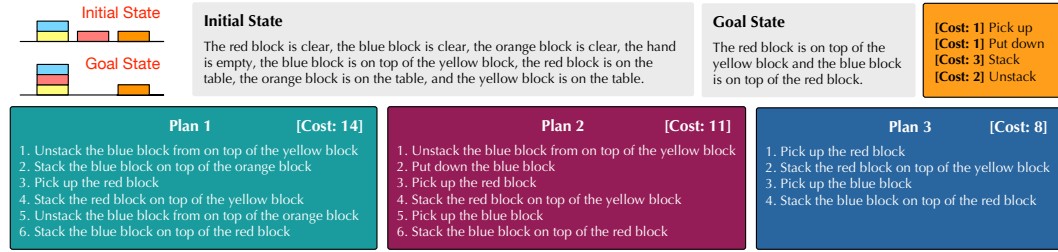

Figure 1: BlocksWorld includes four actions, *pick up, put down, stack, and unstack*, each with an associated cost. Plans that are optimal in terms of execution steps may not be cost-optimal. E.g., two plans with the same number of steps (6) can have different total costs (Plan 1 costs 14, while Plan 2 costs 11). Plans directly generated by LLMs may contain hallucinations that lead to invalid actions. For instance, in Plan 3, the model attempts to stack the red block on the yellow block, even though the blue block is already on top of the yellow block.

Search algorithms are foundational to sequential decision-making, providing a principled mechanism for looking ahead, comparing alternatives, and managing uncertainty. Combining LLM with single-tree-based search algorithms has been proven effectively on sequential decision making tasks (Hao et al., 2023; Zhao et al., 2023; Gao et al., 2024; Lehnert et al., 2024). However, single-tree-based planners face a key limitation: the search space grows exponentially with depth, leading to poor performance on long-horizon tasks. To address this, we introduce a cost-augmented bidirectional search (CATS) that expands two trees, one forward tree starting search from the initial state and one backward tree starting from the goal, coupled by a cross-tree guidance signal. The search halts when the trees meet at a shared state, at which point the full plan is constructed. CATS incorporate cost-awareness at every step: actions are weighted by expense, and task constraints (e.g., "complete within $100") prune unfeasible paths. Our experimental results reveal that CATS outperforms both raw LLM planners and single-tree-based planners, achieving 100% success rate on unlimited budget setting. Having outlined the testbed and design of CATS, we now summarize our key contributions:

- **Cost-Augmented Benchmark** We introduce Budget-BlocksWorld, a variant of the classic BlocksWorld benchmark augmented with non-uniform action costs and configurable budget constraints, transforming a step-counting task into a cost-sensitive planning benchmark.

- **Cost-Aware Evaluation** We propose a comprehensive evaluation suite consisting of *Success Rate*, *Optimality*, and *Efficiency*, which together assess planners on budget-constrained tasks in terms of solution correctness, cost-effectiveness, and search efficiency.

- **Cost-Augmented Tree Search** We introduce **CATS**, a novel tree-based search algorithm that grows two trees, one forward from the initial state and one backward from the goal, coupled through a cross-tree guidance signal. Experiments show that CATS outperforms classic search methods on all tasks in Budget-BlocksWorld.

## 2 RELATED WORKS

### 2.1 TEST TIME COMPUTING

Test-time compute has emerged as a key paradigm for enhancing post-training reasoning. AlphaGo Zero achieved superhuman performance by leveraging Monte Carlo Tree Search (MCTS) with substantial test-time compute (Silver et al., 2016). Similarly, recent LLMs like o1 (Jaech et al., 2024) show that increasing inference-time computation can significantly enhance reasoning. In practice, test time compute is initiated trough repeated sampling, self correction, tree search (Wang et al., 2022; Shinn et al., 2023; Snell et al., 2024; Kumar et al., 2024; Yang et al., 2024). Repeated sampling improves reliability by generating multiple candidate solutions and selecting the most consistent answer, as in self-consistency decoding. Self-correction instead refines an initial response by prompting the model to re-evaluate and revise its outputs, often using external feedback or verification signals. However, both approaches operate at the level of isolated answers and are generally incapable of handling multi-step planning tasks that require explicit state transitions and

structured search. In contrast, tree-based search algorithms have gained traction because it allocates test-time compute toward structured exploration, enabling LLMs to reason over multi-step planning tasks with explicit state transitions.

## 2.2 Tree-based Search algorithms in LLM era

LLMs such as ChatGPT (Brown et al., 2020) exhibit strong commonsense reasoning, extending planning beyond purely symbolic domains to open-ended tasks (Xie et al., 2024; Jimenez et al., 2023; Hausknecht et al., 2019), requiring commonsense reasoning. However, direct calls to an LLM depend solely on its internal algorithm, where state transitions are handled inherently but remain opaque and error-prone. In contrast, external algorithms such as tree search or other robust search procedures define the environment explicitly, enabling systematic exploration and verifiable state transitions. This motivates coupling LLMs with tree-based search algorithms for planning tasks. Several studies have employed tree search during inference time to guide the LLM in sequential decision-making tasks (Zhang et al., 2023; Feng et al., 2023; Hu et al., 2024; Besta et al., 2024; Chen et al., 2024; Antoniades et al., 2024; Shi et al., 2025; Zhang et al., 2025b) . Beyond inference, tree-based search has also been leveraged to generate reasoning trajectories that serve as training data for improving downstream models (Mittal et al., 2025; Zhang et al., 2024; Light et al., 2025). All of those methods construct a single search tree rooted at the initial state, causing the search space to grow exponentially with depth. Our method explores the path from both initial and goal state in order to reduce the depth of each search tree. This bidirectional design not only mitigates the exponential blowup in search depth but also enables more efficient convergence between two trees.

## 2.3 Cost-augmented planning problem

Planning is a fundamental aspect of human intelligence, involving the generation of action sequences to search for solutions and make decisions (Hayes-Roth & Hayes-Roth, 1979; Mattar & Lengyel, 2022; Su, 2023). This capability has been extensively studied in domains such as robotics and autonomous systems (Alterovitz et al., 2016; McDermott, 1992). Recent advances have shown that LLMs and LLM-based frameworks can achieve remarkable performance on a variety of planning tasks. This success is largely due to their strong general reasoning abilities and their integration with classical data structures such as trees and graphs (Hao et al., 2023; Yao et al., 2023; Besta et al., 2024). However, most existing planning benchmarks like GSM8K (Cobbe et al., 2021), Game of 24, judge performance solely on plan correctness, largely ignoring real-world constraints such as time, cost, or resource limits. In practical applications, planning often operates under strict budget constraints, such as limited time steps, energy consumption, or financial cost, making it essential to generate not only valid but also efficient plans (Xie et al., 2024; Zhang et al., 2025a). In this work, we introduce a cost-aware concept to the BlocksWorld benchmark that explicitly incorporates budget constraints into the planning process.

## 3 Budget-BlocksWorld

We introduce our testbed Budget-BlocksWorld, a cost-augmented version of BlocksWorld benchmark (Valmeekam et al., 2022). BlocksWorld is a widely used planning benchmark with several distinct blocks on a table. The objective of this task is to rearrange the blocks on the table from its initial state to the goal state using four deterministic actions: pick-up, put-down, stack, and unstack. Building on top of this setup, Budget-BlocksWorld obtains 1,008 tasks with 6-block instances, and categorizes the difficulty by the optimal plan length (minimum number of actions from initial state to goal). We split our dataset into three categories: **short-horizon** (2–8 steps), **mid-horizon** (10–14 steps), and **long-horizon** (16+ steps) to capture different levels of planning difficulty, where longer horizons require reasoning over more complex state transitions and larger search space. We adopt BlocksWorld with action costs to transform it into Budget-BlocksWorld, a realistic cost-sensitive planning task that is unique among conventional planning benchmarks. We assign non-uniform costs to actions, making some actions (e.g., put-down/pick-up) more expensive than others (e.g., stack/unstack), which forces planners to trade off plan length against execution cost.

| Action costs (pu, un, pd, st) | Short-Horizon | Mid-Horizon | Long-Horizon | Total |
|---|---|---|---|---|
| $[20, 1, 1, 1]$ | 80% | 79% | 95% | 84% |
| $[1, 20, 1, 1]$ | 81% | 68% | 76% | 71% |
| $[1, 1, 20, 1]$ | 86% | 86% | 96% | **90%** |
| $[1, 1, 1, 20]$ | 74% | 62% | 75% | 67% |
| $[20, 1, 20, 1]$ | 62% | 83% | 97% | 87% |
| $[1, 20, 1, 20]$ | 21% | 58% | 79% | 58% |
| $[20, 20, 1, 1]$ | 48% | 67% | 85% | 69% |
| $[1, 1, 20, 20]$ | 44% | 66% | 83% | 67% |
| $[20, 1, 1, 20]$ | 26% | 58% | 80% | 59% |
| $[1, 20, 20, 1]$ | 53% | 71% | 75% | 69% |

Table 1: **Solution shift rate under nonuniform action costs.** The first column represents the cost schedule allocated for each action: pick-up(pu), unstack(un), put-down(pd), stack(st). An optimal solution is counted as *shifted* if its *sequence of action types* under the given schedule is not exactly the same as under the uniform-cost solution. The rest columns represent the percentage of tasks whose optimal solutions are shifted.

**Ground Truth Generation.** To establish the cost distribution for our experiments, we evaluate several cost schedules and recompute ground-truth plans to ensure cost optimality. Using exhaustive search, we enumerate all possible plans and select the lowest-cost plan as the new ground truth.

**Solution Shift Rate.** We then compute the solution shift rate for each cost schedule. This metric measures the proportion of tasks in Budget-BlocksWorld whose cost-optimal plans differ from the uniform-cost BlocksWorld baseline. A task's solution is considered as shifted if the cost-optimal ground truth plan is not identical to the uniform ground truth plan. Table 1 summarizes these results. We select the [1, 1, 20, 1] configuration with the highest ratio of shifted plans as our testbed, ensuring that our optimal solutions are clearly distinguishable from those of the vanilla BlocksWorld. This setup emphasizes the robustness of planning algorithms by requiring them to optimize not only for plan feasibility but also for execution cost under varying action-cost configurations. We provide more statistics on our cost-augmented BlocksWorld in Appendix A.

Budget-BlocksWorld has three budget conditions to simulate different levels of cost strictness. TIGHT budget is set exactly to the ground-truth cost-optimal plan, permitting only plans whose cost matches the optimum. LOOSE budget is set to the optimal cost plus a small margin, allowing slight overruns. We set the small margin to be the cost of two rounds of most expensive actions. UNLIMITED condition applies no budget constraints, and all valid plans are accepted. This setup allows us to systematically study planner's behavior under strict, relaxed, and unconstrained cost settings.

## 4 COST-AUGMENTED TREE SEARCH (CATS)

We introduce CATS to address budget-constrained planning problems. In the Budget-BlocksWorld setting, the planner is given an initial block state $s_0$, where uniquely colored blocks are stacked on the table in a specific order (Figure 1). The planner is also provided with a budget limit $\mathcal{B}$, an action set $\mathcal{A}$, and a goal state $s_g$. Each action $a \in \mathcal{A}$ (e.g., picking up the blue block; stacking the red block on the top of the green block) is associated with a cost $c_a$. The planner's objective is to select a sequence of actions $\mathcal{A}' = [a_1, a_2, \ldots, a_k]$ from the action set $\mathcal{A}$ that transforms the initial state $s_0$ into the goal state $s_{goal}$ when applied in order. More importantly, the total cost incurred by executing $\mathcal{A}'$, i.e., $\sum_{i=1}^{k} c_{a_i}$, must not exceed the budget limit $\mathcal{B}$.

CATS adopts a tree search methodology to solve the Budget-BlocksWorld task. Given the initial state $s_0$ and the goal state $s_{goal}$, CATS constructs a forward search tree $T_f$ rooted at node $v_0$, and a backward search tree $T_b$ rooted at node $v_{goal}$. The two trees expand in alternating turns, each expanding one selected leaf node per round by applying actions $a \in \mathcal{A}$ as edges. Each node $v$ is associated with a cost $c_v$, representing the cumulative edge cost along the path from the corresponding root, either $v_0$ in the forward tree or $v_{goal}$ in the backward tree, to $v$. In this way, CATS transforms the challenging problem of finding sequence $\mathcal{A}'$ into building a valid path between $v_0$ and $v_{goal}$ via growing two trees. To prevent non-termination, we impose a node expansion limit $L$. If the two trees

Figure 2: An example of CATS. (1) Forward tree selects the highest reward node $v_1$ to expand. (2) Forward tree checks if it has nodes equivalent to a node in the backward tree. (3) Backward tree selects node $v_4$ to expand. (4) Backward tree expands and finds $v_8$ equals to the $v_6$ in the forward tree, completes the search. *Non-essential comparisons edges omitted; search result remains valid.*

fail to meet within $L$ expansions, CATS halts the search and declares that no valid path is found. We provide an example of CATS building such path in Figure 2.

We design a reward mechanism that guides the selection of the most promising node in $T_f$ and $T_b$ for expansion. We leverage the reasoning capabilities of LLMs to construct a *confidence reward*, which quantifies the utility of applying an action $a$ at a given node $v$. Additionally, we introduce a *distance reward*, based on Jaccard distance(Manning et al., 2008), to encourage efficient convergence between the two search trees. We detail the full reward design in Section 4.1.

Section 4.2 details how CATS selects nodes for expansion based on its reward functions and incrementally grows $T_f$ and $T_b$ toward each other. When two equivalent nodes appear in both trees, indicating that a common node $v$ is reachable from both $v_0$ and $v_{goal}$, CATS terminates the expansion process and completes the search. The full procedure is outlined in Algorithm 1.

## 4.1 REWARD DESIGN

To efficiently grow the two search trees toward convergence, CATS employs two reward signals to guide node selection. Each node in $T_f$ and $T_b$, excluding $v_0$ and $v_{goal}$, is assigned a reward score, indicating its likelihood of lying on the valid path connecting the trees. The **confidence reward** $R_{conf}$ assesses the node's role within its own tree, reflecting the planner's internal confidence. The **distance reward** $R_{dist}$ evaluates a node's proximity in jaccard distance to the opposite tree, promoting convergence. Both rewards are normalized and combined using a weighting parameter $\omega \in [0, 1]$ to balance their influence.

**Confidence Reward.** Consider a leaf node $v$ with parent node $v_p$. Let $a$ be the action that transforms $v_p$ into $v$, and let $c_a$ be its cost. The parent node has an accumulated cost $c_{v_p}$. To assess the quality of $a$, we prompt the LLM with the parent node $v_p$, the available action set $\mathcal{A}$, the budget limit $\mathcal{B}$, and the accumulated cost $c_{v_p}$. The LLM is asked to predict the best next action. Since $a$ was the action taken, we use the log-probability that the LLM assigns to $a$ as the confidence reward for node $v$. To reduce potential hallucinations or inconsistencies, we add a second layer of validation using LLM-based self-evaluation. Given the action $a$, the budget limit $\mathcal{B}$, parent node $v_p$ and its cost $c_{v_p}$, we prompt the LLM to assess whether $a$ is a good choice by outputting either *good* or *bad*. The detailed prompts are included in Appendix B. We then use the log-probability of generating *good* as an additional confidence signal for the node $v$. The reward calculation is formalized in Equation 1, and is used by both the forward and backward trees to guide their node selection during the expansion process.

$$R_{conf}(v) = \log P_{\text{LLM}}(a \mid v_p, c_{v_p}, \mathcal{B}) + \log P_{\text{LLM}}(good \mid a, v_p, \mathcal{B}, c_{v_p}) \qquad (1)$$

---

**Algorithm 1** Cost-Augmented Tree Search (CATS)

---

**Require:** Initial node $v_0 \in \mathbb{V}$, goal node $v_g \in \mathbb{V}$, weight $\omega$, max iterations $L$, budget limit $\mathcal{B}$
1: Initialize confidence reward $R_{conf} : \mathbb{V} \to \mathbb{R}$, distance reward $R_{dist} : \mathbb{V} \times \mathbb{V} \to \mathbb{R}$
2: Initialize feasible actions $A : \mathbb{V} \times \mathcal{B} \to \mathbb{A}$, child nodes $c : \mathbb{V} \times \mathbb{A} \to \mathbb{V}$
3: Initialize unvisited verifier $U : \mathbb{V} \to \{0, 1\}$
4: Initialize node set $\mathbb{V}_f \leftarrow \{v_0\}$, $\mathbb{V}_b \leftarrow \{v_g\}$
5: Initialize empty plan $\pi \leftarrow \emptyset$
6: **for** $t \leftarrow 0$ **to** $L - 1$ **do**
7:     $v_f^* \leftarrow \arg\max_{v \in Leaf(v_0)} (\omega \cdot U(v) \cdot R_{conf}(v) + (1 - \omega) \cdot R_{dist}(v))$
8:     **for** $a \in A(v_f^*, \mathcal{B})$ **do**
9:         $v^* \leftarrow c(v_f^*, a)$
10:        $\mathbb{V}_f \leftarrow \mathbb{V}_f \cup \{v^*\}$
11:     **end for**
12:     $v_b^* \leftarrow \arg\max_{v \in Leaf(v_g)} (\omega \cdot U(v) \cdot R_{conf}(v) + (1 - \omega) \cdot R_{dist}(v))$
13:     **for** $a \in A(v_b^*, \mathcal{B})$ **do**
14:         $v^* \leftarrow c(v_b^*, a)$
15:        $\mathbb{V}_b \leftarrow \mathbb{V}_b \cup \{v^*\}$
16:     **end for**
17:     $v_{\text{overlap}} \leftarrow \mathbb{V}_f \cap \mathbb{V}_b$
18:     **if** $v_{\text{overlap}} \neq \emptyset$ **then**
19:        $\pi \leftarrow \text{EXTRACTPLAN}(v_{\text{overlap}})$
20:        **break**
21:     **end if**
22: **end for**
23: **return** $\pi$

---

**Distance Reward.** CATS uses Jaccard distance to guide the convergence of the forward and backward search frontiers. For a leaf node $v$, we compute its distance reward to the opposite tree $T_{opp} \in \{T_f, T_b\}$ as the minimum Jaccard distance which is between $v$ and any leaf node of the opposing tree. The distance reward for leaf node $v$ is defined as:

$$R_{\text{dist}}(v) = 1 - \min_{v_{opp} \in \text{Leaf}(T_{opp})} J(v, v_{opp}) \quad \text{where} \quad J(v, v_{opp}) = \frac{|\mathcal{S}(v) \cap \mathcal{S}(v_{opp})|}{|\mathcal{S}(v) \cup \mathcal{S}(v_{opp})|} \quad (2)$$

where $\mathcal{S}(v)$ denotes the set of block-state descriptions represented by node $v$. For example, in the initial state in Figure1, the statement "the blue block is on top of the yellow block" correspond to a single block-state description.

## 4.2 TREE EXPANSION ALGORITHM

The node selection and expansion process is crucial, as it ensures that CATS focuses its time and budgets on growing the most cost-effective branches of the search trees.

**Selection.** CATS selects the most promising node for expansion based on the *confidence* and *distance* rewards defined in the previous section 4.1. For a search tree $T \in \{T_f, T_b\}$, the selected leaf node $v^* \in \text{Leaf}(T)$ is determined as:

$$v^* = arg \max_{v \in \text{Leaf}(T)} \left[ \omega \cdot U(v) \cdot R_{conf}(v) + (1 - \omega) \cdot R_{dist}(v) \right] \quad (3)$$

$U(v)$ is a binary indicator, where $U(v) = 0$ if the node $v$ has already been visited, and $U(v) = 1$ otherwise. The weight parameter $\omega$ controls the trade-off between the confidence reward and the distance reward.

**Expansion.** After selecting the node $v^*$, CATS generates successor nodes for all feasible actions in $\mathcal{A}$ that can be applied to the current block state of $v^*$. To prevent redundant exploration, it performs a node merging step: if a successor shares the same block state as an existing node in the tree, only the node with the lower path cost is retained. If all successors have higher costs than their matching counterparts, the node $v^*$ is not expanded and is instead marked as visited by setting $U(v) = 0$. This strategy avoids cycles and ensures the search tree grows along cost-effective paths only.

## 5 EXPERIMENTS

We compare CATS against two types of baselines: tree-based planners such as Tree-of-Thought (ToT), Reasoning via Planning(RAP), and state-of-the-art raw LLM planners (GPT-4.1, Claude-Opus-4.1).

**Raw LLM planner.** We directly prompt the model with in-context learning examples and require it to generate the entire plan in one pass. The cost budget limit specified in the prompt is varied according to the given constraint. The full prompting details are included in the Appendix C.

**Tree-based LLM planner.** We compare our method CATS with two well-known tree-based planners, ToT and RAP. Tree of Thought (ToT) extends chain-of-thought prompting, where an LLM reasons step-by-step, by turning these steps into a search tree. Instead of following only one line of reasoning, ToT explores multiple possible reasoning paths using classical search methods like DFS or BFS, and uses the LLM to score which path look more promising. Reasoning via Planning (RAP) goes a step further: instead of only guiding a search with the LLM, it integrates the LLM into Monte Carlo Tree Search (MCTS). MCTS adaptively balances exploration (trying new reasoning paths) and exploitation (refining promising ones), allowing RAP to continuously reconsider alternatives and avoid getting stuck on suboptimal paths. Both ToT and RAP are using same confidence reward mentioned in section 4.1 to guide the tree search. To standardize the implementation and reduce hallucination of LLM in our implementation, we use PDDLGym (Silver & Chitnis, 2020) as the world model to update the state based on the given state-action pair.

Additionally, we introduce a strict bounding mechanism during the tree-search process for ToT and RAP: any action that exceeds the cost budget limit is eliminated, thereby improving overall performance. For a fair comparison, we set a maximum tree node expansion limit of 500 nodes for ToT, RAP and CATS. This prevents the planner from resorting to exhaustive search.

### 5.1 METRICS

We evaluate the performance using three metrics: **Success Rate**, **Optimality**, **Efficiency**. The success rate checks whether the planner find a valid, cost-adherent plan within the node expansion limits. Optimality quantifies cost efficiency by comparing the generated plan's cost to that of the ground-truth optimal plan. We compute the cost difference between the two and apply a reciprocal transformation that maps smaller deviations to larger scores. The resulting optimality score ranges from 0 to 0.5. In this way, the optimality metric is defined such that higher scores indicate better performance (see Eq. 4). Efficiency reflects the number of node expansion of the search tree required to obtain the first feasible plan. A higher efficiency value means the planner reached a solution while consuming a smaller portion of its allotted node expansion budget (see Eq. 4).

$$\text{Optimality} = \frac{1}{1 + (\text{Cost}_{gen}/\text{Cost}_{opt})}; \quad \text{Efficiency} = 1 - \frac{\#\,\text{Nodes Expanded}}{\text{Node Budget}} \qquad (4)$$

### 5.2 OVERALL PERFORMANCE

Table 2 compares Success Rate and Optimality metrics across three budget limits (TIGHT, LOOSE, UNLIMITED) and plan lengths (Short, Mid, Long). Under LOOSE budget, CATS attains a success rate of 84% on mid-horizon tasks and maintains 36% on long-horizon tasks. With the UNLIMITED budget, it further achieves an overall 100% success rate. Besides substantial gains in success rate, CATS consistently achieves the highest optimality across all budget settings. Under the LOOSE and UNLIMITED settings, it achieves average optimality scores of 0.27 and 0.33, more than twice the performance of Claude-Opus-4.1, the second-best performing method. This demonstrates the effectiveness of CATS's core design, which integrates LLM-based reasoning with structured, cost-aware search.

Search algorithms can substantially enhance the performance of pure LLM planners, but their effectiveness varies widely. While ToT and RAP improve Qwen3-8B over plain CoT, they still fail to surpass CoT with stronger closed-source models such as GPT-4.1 or Claude-Opus-4.1, and their gains are often confined to short-horizon tasks. In contrast, CATS consistently outperforms all other search strategies across both budget constraints and task horizons. It achieves an average success rate of 65% under the LOOSE setting and 100% under the UNLIMITED budget, compared to just 2%

| | PLAN LENGTH | Success Rate | | | | Optimality | | | |
|---|---|---|---|---|---|---|---|---|---|
| | | Short | Mid | Long | Avg. | Short | Mid | Long | Avg. |
| TIGHT | CoT w/ Qwen3-8B | 0.04 | 0.00 | 0.00 | 0.01 | 0.04 | 0.00 | 0.00 | 0.01 |
| | CoT w/ GPT-4.1 | 0.04 | 0.01 | 0.00 | 0.01 | 0.04 | 0.01 | 0.00 | 0.01 |
| | CoT w/ Claude-Opus-4.1 | 0.19 | 0.01 | 0.00 | 0.02 | 0.16 | 0.01 | 0.00 | 0.02 |
| | ToT w/ Qwen3-8B | 0.08 | 0.00 | 0.00 | 0.01 | 0.04 | 0.00 | 0.00 | 0.01 |
| | RAP w/ Qwen3-8B | 0.34 | 0.01 | 0.00 | 0.04 | 0.16 | 0.01 | 0.00 | 0.02 |
| | **CATS w/ Qwen-8B** | **0.34** | **0.08** | **0.01** | **0.08** | **0.16** | **0.04** | **0.01** | **0.04** |
| LOOSE | CoT w/ Qwen3-8B | 0.04 | 0.00 | 0.00 | 0.01 | 0.04 | 0.00 | 0.00 | 0.01 |
| | CoT w/ GPT-4.1 | 0.21 | 0.15 | 0.05 | 0.11 | 0.05 | 0.05 | 0.02 | 0.04 |
| | CoT w/ Claude-Opus-4.1 | 0.58 | 0.25 | 0.14 | 0.24 | 0.20 | 0.09 | 0.05 | 0.08 |
| | ToT w/ Qwen3-8B | 0.14 | 0.00 | 0.00 | 0.02 | 0.06 | 0.00 | 0.00 | 0.02 |
| | RAP w/ Qwen3-8B | 0.61 | 0.04 | 0.01 | 0.09 | 0.21 | 0.01 | 0.01 | 0.03 |
| | **CATS w/ Qwen-8B** | **0.96** | **0.84** | **0.36** | **0.65** | **0.31** | **0.29** | **0.24** | **0.27** |
| UNLIMITED | CoT w/ Qwen3-8B | 0.04 | 0.00 | 0.00 | 0.01 | 0.04 | 0.01 | 0.01 | 0.01 |
| | CoT w/ GPT-4.1 | 0.27 | 0.19 | 0.10 | 0.16 | 0.07 | 0.06 | 0.03 | 0.05 |
| | CoT w/ Claude-Opus-4.1 | 0.55 | 0.32 | 0.23 | 0.31 | 0.18 | 0.10 | 0.07 | 0.10 |
| | ToT w/ Qwen3-8B | 0.15 | 0.02 | 0.00 | 0.02 | 0.06 | 0.01 | 0.00 | 0.01 |
| | RAP w/ Qwen3-8B | 0.69 | 0.04 | 0.01 | 0.10 | 0.25 | 0.01 | 0.00 | 0.03 |
| | **CATS w/ Qwen-8B** | **1.00** | **1.00** | **1.00** | **1.00** | **0.29** | **0.32** | **0.34** | **0.33** |

Table 2: Overall performance of planners on Budget-BlocksWorld. CATS consistently outperforms all other methods across budget conditions and task horizons. The comparison includes raw LLM planners (CoT w/ model) and tree-based LLM planners (ToT, RAP, CATS). For CATS, we set $\omega = 0.5$ in the selection schema (Equation 3). In the LOOSE setting, we provide an additional budget of 42, corresponding to the cost of two full rounds of pick-up/unstack followed by put-down actions.

| | PLAN LENGTH | Short | Mid | Long | Avg. |
|---|---|---|---|---|---|
| TIGHT | ToT w/ Qwen3-8B | 0.92 | n/a | n/a | 0.92 |
| | RAP w/ Qwen3-8B | 0.90 | 0.50 | n/a | 0.85 |
| | **CATS w/ Qwen-8B** | **0.99** | **0.97** | **0.97** | **0.98** |
| LOOSE | ToT w/ Qwen3-8B | 0.92 | n/a | n/a | 0.92 |
| | RAP w/ Qwen3-8B | 0.77 | 0.48 | 0.71 | 0.71 |
| | **CATS w/ Qwen-8B** | **0.97** | **0.94** | **0.92** | **0.93** |
| UNLIMITED | ToT w/ Qwen3-8B | 0.92 | 0.93 | n/a | 0.92 |
| | RAP w/ Qwen3-8B | 0.76 | 0.74 | n/a | 0.70 |
| | **CATS w/ Qwen-8B** | **0.97** | **0.93** | **0.91** | **0.93** |

Table 3: The efficiency of tree-based LLM planners across different level of horizon tasks and constraint settings. "n/a" indicates no successful tasks in that category.

and 9% for ToT and RAP, respectively, under LOOSE. These results highlight that CATS uniquely enables small open-source models to surpass state-of-the-art CoT baselines by using bidirectional search and cross-tree guidance.

## 5.3 EFFICIENCY

In this section, we compare efficiency across methods. A higher efficiency score indicates less node expansion usage during the tree seach process. As shown in Table 3, **CATS** achieves the highest average efficiency across all budget settings, reaching 98% under TIGHT constraints compared to 92% for ToT. This demonstrates that CATS is particularly effective when strict budget adherence is required. Across plan lengths, efficiency decreases monotonically for all methods, but CATS remains the most consistent, with its largest drop only 6% from short- to long-horizon in the unlimited setting, while ToT often fails on long-horizon tasks. These results show that CATS reduces nearly 90% of unnecessary expansions compared to cost-agnostic search, demonstrating that our reward design effectively guides the trees toward the correct path for convergence.

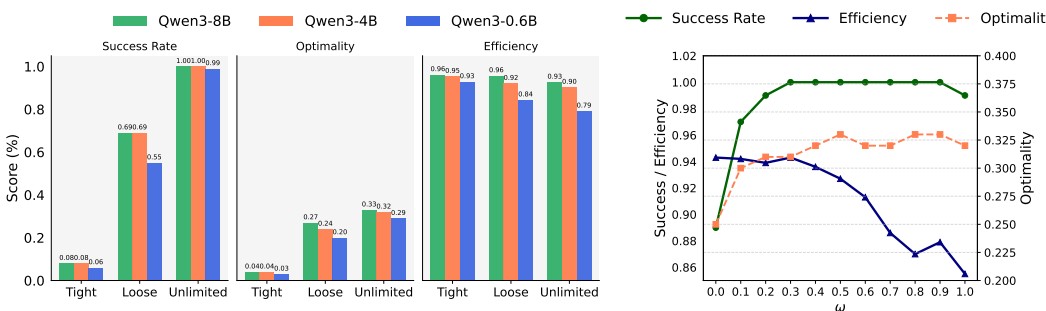

Figure 3: Performance comparison across models (left). CATS is effective across different scales of model, while larger model intuitively provides stronger performance. Performance across different $\omega$ values (right): Balancing confidence and distance rewards yields the best overall performance.

## 5.4 ABLATION STUDIES

**Model Size.** To examine the robustness of CATS across model scales, we evaluate it with Qwen3-0.6B, Qwen3-4B, and Qwen3-8B. All other variables are held fixed, with $\omega = 0.5$, and experiments are conducted under TIGHT, LOOSE, and UNLIMITED constraint settings. From the bar chart in Figure 3, we can see that the results for the three models are broadly consistent: larger models generally yield stronger performance. Qwen3-4B and Qwen3-8B has relatively same performance while Qwen3-0.6B is less cost-efficient compared to its other models with around 14% drop in efficiency under UMLIMITED settings compare to Qwen3-8B. These findings suggest that CATS is robust across model scales, with larger models offering incremental benefits but not altering the overall effectiveness of the method. Remarkably, even Qwen3-0.6B surpasses the latest state-of-the-art LLM, ToT and RAP planner across all constraint settings.

**Distance Reward.** In this section, we investigate how distance reward influence the reward signal of CATS. Specifically, we vary the weighting parameter $\omega$ in Equation 3 from 0 to 1 in increments of 0.1. Setting $\omega = 1$ corresponds to relying solely on the *confidence reward*, whereas $\omega = 0$ represents the reward is entirely depended on the *distance reward*. We run the experiment using Qwen3-8B as the base model, and assess performance on the entire Budget-BlocksWorld with UNLIMITED constraint. As shown in line chart in Figure 3, three key observations emerge: (1) success rate is highly robust across a wide range of $\omega$, achieving nearly perfect performance when $\omega \geq 0.3$. (2) Optimality generally improves as $\omega$ increases, showing that solutions become more cost-efficient when the *confidence reward* signal is given greater weight, while heavy reliance on *distance reward* alone reduces plan quality. (3) Efficiency initially benefits from incorporating *distance reward* but declines when $\omega$ becomes too large, with performance degrading when rewards rely solely on distance reward. These results suggest that a balanced combination of *confidence reward* and *distance reward* yields the most efficient outcomes while maintaining high success rates and optimality.

## 6 CONCLUSION

We conducted a systematic study of LLM-assisted cost-sensitive planning. To fill the gap in evaluating cost-sensitive planning, we proposed Budget-Blocksworld, a cost-augmented benchmark, along with a complementary evaluation suite that measures plan success, cost efficiency, and search effectiveness. We introduced a novel method, **CATS**, which integrates cost-awareness into LLM-guided search. While raw LLMs such as GPT-4.1 often fail under tight budgets, CATS consistently generates cost-efficient plans by combining structured search with LLM reasoning. Notably, CATS achieves state-of-the-art performance on Budget-BlocksWorld, delivering the highest success rates, optimality and search efficiency, across all budget constraints and long-horizon tasks. Our work suggests several promising directions for future work, including extending CATS to domains with richer constraints and integrating reinforcement learning to enable planners that internalize cost-augmented reasoning.

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

## A  COST-AUGMENTED BLOCKSWORLD

The Budget-BlocksWorld dataset contains tasks of varying lengths, from 2 to 32 steps. The distribution is as follows: 2-step (5), 4-step (12), 6-step (29), 8-step (67), 10-step (108), 12-step (177), 14-step (191), 16-step (163), 18-step (109), 20-step (78), 22-step (33), 24-step (20), 26-step (7), 28-step (6), 30-step (2), and 32-step (1).

We recompute the cost-optimal solution for each task in Budget-BlocksWorld using exhaustive search. At each step, we expand nodes with all feasible actions and retain the least-cost path once a goal state is reached. This non-trivial effort ensures that our benchmark provides reliable optimal ground-truth plans.

## B  ADDITIONAL IMPLEMENTATION DETAILS OF CATS

Figure 4 and Figure 5 displays the prompt used by CATS for generating confidence rewards, and Figure 6 and Figure 7 shows the associated self-evaluation prompt that performs binary validation of each selected action. For consistency, the same prompts are also used in the RAP and ToT baselines.

---

**Next Action Confidence Prompt**

I am playing with a set of blocks where I need to arrange the blocks into stacks. Here are the actions I can do and the corresponding time consumption:
Pick up a block, 1 min
Unstack a block from on top of another block, 1 min
Put down a block, 20 min
Stack a block on top of another block, 1 min
I have the following restrictions on my actions:
I can only pick up or unstack one block at a time.
I can only pick up or unstack a block if my hand is empty.
I can only pick up a block if the block is on the table and the block is clear. A block is clear if the block has no other blocks on top of it and if the block is not picked up.
I can only unstack a block from on top of another block if the block I am unstacking was really on top of the other block.
I can only unstack a block from on top of another block if the block
I am unstacking is clear. Once I pick up or unstack a block, I am holding the block.
I can only put down a block that I am holding.
I can only stack a block on top of another block if I am holding the block being stacked.
I can only stack a block on top of another block if the block onto which I am stacking the block is clear.
Once I put down or stack a block, my hand becomes empty.

[STATEMENT] As initial conditions I have that, the red block is clear, the blue block is clear, the orange block is clear, the hand is empty, the red block is on the table, the blue block is on the table and the orange block is on the table.
My goal is to have that the blue block is on top of the orange block.
My previous plan is:
[PREVIOUS PLAN] pick up the blue block [PREVIOUS PLAN END]
The current time usage for the previous plan is 1 min
My plan is as follows:
[PLAN] stack the blue block on top of the orange block [PLAN END]

---

Figure 4: The prompt for computing the confidence reward for CATS.

---

**Next Action Confidence Prompt(Continued)**

[Another In-context learning example in the same format as the previous text block...]

[STATEMENT]
As initial conditions I have that, <init state>
My goal is to <goals>
My previous plan is:
[PREVIOUS PLAN] <previous plan> [PREVIOUS PLAN END]
The current time usage for the previous plan is <current time usage> min
My plan is as follows:
[PLAN]

---

Figure 5: The prompt for computing the confidence reward for CATS.

---

**Self Evaluation Prompt for Confidence Reward**

I am playing with a set of blocks where I need to arrange the blocks into stacks. Here are the actions I can do and the corresponding time consumption:
Pick up a block, 1 min
Unstack a block from on top of another block, 1 min
Put down a block, 20 min
Stack a block on top of another block, 1 min
I have the following restrictions on my actions:
I can only pick up or unstack one block at a time.
I can only pick up or unstack a block if my hand is empty.
I can only pick up a block if the block is on the table and the block is clear. A block is clear if the block has no other blocks on top of it and if the block is not picked up.
I can only unstack a block from on top of another block if the block I am unstacking was really on top of the other block.
I can only unstack a block from on top of another block if the block
I am unstacking is clear. Once I pick up or unstack a block, I am holding the block.
I can only put down a block that I am holding.
I can only stack a block on top of another block if I am holding the block being stacked.
I can only stack a block on top of another block if the block onto which I am stacking the block is clear.
Once I put down or stack a block, my hand becomes empty.

[STATEMENT] The initial state: I have that, the yellow block is on the table. The red block is on the top of the yellow block. The green block is on the top of the red block. The blue block is on the green block. The blue block is clear. The hand is clear.
The current state: I have that, the yellow block is on the table. The red block is on the top of the yellow block. The green block is on the top of the red block. The blue block is on the table. The blue block is clear. The hand is clear.
Goal: The green block is on the top of blue block.
[PREVIOUS ACTION]
unstack the blue block from the green block.
put down the blue block.
We have used 21 minutes.
Our time limit is 33 minutes
[ACTION] unstack the green block from the red block
[EVALUATION] good

[Another in-context learning example in the same format as the previous text block...]

---

Figure 6: The prompt for computing the confidence reward for CATS.

---

**Self Evaluation Prompt for Confidence Reward (Continued)**

[STATEMENT]
The initial state: I have that, <init state>
The current state: I have that,<current state>
Goal: <goal>
[PREVIOUS ACTION] <previous actions>
We have used <current time usage> minutes so far.
Our time limit is <time limit> minutes.
[ACTION] <action>
[EVALUATION]

---

Figure 7: The prompt for computing the confidence reward for CATS

## C   BASELINE CONFIGURATIONS

Figure 8 and Figure 9 illustrates the prompt employed for GPT-4.1, Claude-Opus-4.1, and Qwen3-8B in our main experiments (Section 5.2).

---

**Prompt for raw LLM planner**

I am playing with a set of blocks where I need to arrange the blocks into stacks. Here are the actions I can do and the corresponding time consumption:
Pick up a block, 1 min
Unstack a block from on top of another block, 1 min
Put down a block, 20 min
Stack a block on top of another block, 1 min
I have the following restrictions on my actions:
I can only pick up or unstack one block at a time.
I can only pick up or unstack a block if my hand is empty.
I can only pick up a block if the block is on the table and the block is clear. A block is clear if the block has no other blocks on top of it and if the block is not picked up.
I can only unstack a block from on top of another block if the block I am unstacking was really on top of the other block.
I can only unstack a block from on top of another block if the block
I am unstacking is clear. Once I pick up or unstack a block, I am holding the block.
I can only put down a block that I am holding.
I can only stack a block on top of another block if I am holding the block being stacked.
I can only stack a block on top of another block if the block onto which I am stacking the block is clear.
Once I put down or stack a block, my hand becomes empty.

IMPORTANT: You must respond with ONLY the plan in the exact format shown below. Do not include any explanations, analysis, or additional text.

[STATEMENT] As initial conditions I have that, the red block is clear, the blue block is clear, the orange block is clear, the hand is empty, the red block is on the table, the blue block is on the table and the orange block is on the table.
My goal is to have that the blue block is on top of the orange block.
The time limit is 22 min
My plan is as follows:
[PLAN]
pick up the blue block
stack the blue block on top of the orange block
[PLAN END]

---

Figure 8: The prompt for raw LLMs.

---

**Prompt for raw LLM planner (Continued)**

[STATEMENT] As initial conditions I have that, the red block is clear, the blue block is clear, the orange block is clear, the hand is empty, the red block is on top of the yellow block, the blue block is on the table, the orange block is on the table and the yellow block is on the table.
My goal is to have that the blue block is on top of the yellow block and the orange block is on top of the blue block.
The time limit is 45 min
My plan is as follows:
[PLAN]
unstack the red block from on top of the yellow block
put down the red block
pick up the blue block
stack the blue block on top of the yellow block
pick up the orange block
stack the orange block on top of the blue block
[PLAN END]

[STATEMENT] As initial conditions I have that, <init state>
My goal is to <goals>
The time limit is <time limit> min
My plan is as follows:
[PLAN]

---

Figure 9: The prompt for raw LLMs.

## D  USE OF LLMS

We used Large Language Models solely as writing assistants to polish the clarity, grammar, and readability of the manuscript. LLMs were not involved in the research ideation, methodology design, experiment execution, data analysis, or interpretation of results. All technical content, contributions, and conclusions are entirely the work of the authors. The authors take full responsibility for the content of this paper.

