# OpenReview forum: "CATS: Cost-Augmented Tree Search for LLM-Assisted Planning"
_ICLR.cc/2026/Conference — Submitted to ICLR 2026_

### Official Review · Reviewer_2WBo · 2025-10-30

**Soundness:** 2
**Presentation:** 3
**Contribution:** 1
**Rating:** 2
**Confidence:** 4

**Summary:**

The paper introduces CATS (Cost-Augmented Tree Search), an algorithm for LLM-assisted, cost-sensitive planning. The authors identify two main challenges: planning where actions have non-uniform costs and planning under a hard budget constraint (i.e., total cost of operations, which in this paper is the node expansion limit). To address this, the authors make three contributions: A new benchmark of the classical Blocks World domain where actions have different costs; a new evaluation suite with metrics, primarily for hard budget constraints; the CATS algorithm, which is a bidirectional tree search that expands a "forward" tree from the initial state and a "backward" tree from the goal state. The search is guided by an LLM-based "confidence reward" and a Jaccard-based "distance reward" to encourage the two trees to meet. The authors show CATS outperforms baselines by a large margin.

**Strengths:**

* Problem Motivation: The paper identifies a relevant gap in LLM planning research: cost-sensitivity. Most current LLM planning works assume uniform action costs, which is unrealistic for many real-world applications.
* Algorithm Idea: The core idea of using a bidirectional search to reduce the effective planning horizon (and thus the exponential search cost) is a reasonable and established approach in classical search. Applying it to LLM-guided search is a plausible research direction.

**Weaknesses:**

* Weak Motivation: Cost-sensitive planning is not a new problem, it's been widely studied in many planning domains, like traveling salesman problem (TSP). Take TSP as an example, visiting different cities will result in different cost and enumerating all possible solutions will result in exponential time and people have tried using many different heuristics (prioritize search), and even LLM. It's not very clear the profoundness of the first two claimed contribution of the work.

* Weak Benchmarks: The authors' chosen benchmark (Budget-Blocks World) is a variant of a PDDL domain. The authors state in Appendix A that they found the optimal ground-truth plans using *exhaustive search*. This proves the domain is trivially small for classical methods. Given the triviality of the test domain, a simple A* search using the LLM as a heuristic (to estimate $h(n)$) would be a far more relevant and powerful baseline. It is highly probable that a standard heuristic planner would solve 100% of these "long-horizon" (16+ step) tasks in milliseconds, and be far more time-efficient considering the latency of LLM query. The authors should elaborate more on future potential of this line of work, like towards open-world and open-ended planning to strengthen the motivation as well.

* Limited Applicability of Backward Search: The choice of Blocks World is convenient because it has easily reversible actions and a clearly defined, concrete goal state. The authors ignore the fact that bidirectional search is non-trivial (or impossible) for a vast class of open-world reasoning problems, where LLM excels and has significant potential. For example, how is the backward tree constructed for goals like "write a program that passes these test cases" or "discover a new drug molecule"?

* Evaluation suite lack of contribution: The criteria of 'strict, losse, unlimited' corresponds to reporting the distribution of # node expansion on three threshold and defining optimality on [0, 0.5] value is not widely accepted.

* Lack of Baselines and Ablations: The paper's primary claim is that CATS is a superior planning algorithm. However, the CATS algorithm differs from the baselines (ToT, RAP) in two key ways simultaneously:
  * Search Strategy: CATS is bidirectional, while ToT/RAP are single-tree.
  * Reward Signal: CATS uses a confidence reward + distance reward ($R_{conf} + R_{dist}$), while the baselines are stated to use only the confidence reward ($R_{conf}$).

The experiments compare (`Bidirectional + R_conf + R_dist`) to (`Single-Tree + R_conf`). The observed performance gain could come from the bidirectional search, the distance reward, or both. The authors fail to provide the necessary ablation studies to disentangle these factors. The critical missing comparisons are:

  * To test the bidirectional design: CATS (Single-Tree) vs. CATS (Bidirectional).
  * To test the distance reward: RAP (Bidirectional + $R_{conf}$) vs. CATS (Bidirectional + $R_{conf} + R_{dist}$)

**Questions:**

* Backward Tree Construction: In Section 4, the authors state the backward tree expands by "applying actions $a \in \mathcal{A}$." For a correct bidirectional search, the backward tree must apply the inverse of the actions (e.g., `unstack` is the inverse of `stack`) in order to transition from the goal node to its child node. I would like to double check with the authors this is merely a typo, but not present in the experiment. If so, finding and defining inverse actions for domains more complex than Blocks World is not trivial, and may limit the generality of CATS.

* Jaccard Distance Computation: Jaccard distance is used as distance reward, which is easy to use and intuitive. Can you please clarify how the Jaccard distance is computed? Section 4.1 defines $\mathcal{S}(v)$ as the "set of block-state descriptions." For top-down-stack `red-yellow-green`, are its distances with `blue-red-yellow-green` and `red-yellow-green-blue` the same?

* Baseline Failures (Table 3): The "n/a" results for ToT and RAP on mid/long-horizon tasks are attributed to "no successful tasks."
  * Does this mean they failed to find any solution within the 500-node limit?
  * Also it's weired for RAP in long-horizon category, where it succeeds in LOOSE setting but not in UNLIMITED setting. It is expected that the success rate monotonically increase when going from TIGHT, LOOSE to UNLIMITED. And one explanation for the reported result is high randomness. Can authors elaborate on evaluation protocol, like how many rounds are tested and provide an explanation for the result?
  * This loops back to the main weakness: How do we know these failures are due to the single-tree design rather than the lack of your $R_{dist}$ reward?

* See discussion in weakness.

---

### Official Review · Reviewer_E9WQ · 2025-10-31

**Soundness:** 3
**Presentation:** 1
**Contribution:** 1
**Rating:** 0
**Confidence:** 5

**Summary:**

The paper studies the problem of planning with action costs. In particular, the paper focuses on the cost-bounded case, where plans have a maximum cost bound. The authors propose using a classic bi-directional search algorithm to solve the problem, which helps. They evaluate their approach on a modified version of Blocksworld, guiding the search with a domain-dependent heuristic. This has better performance than other LLM-baselines.

**Strengths:**

- The text is clear
- The experimental results are good (with caveats)
- The overall idea is clearly presented, and the figures and plot are informative

**Weaknesses:**

My main issue with the paper is that the main ideas and contributions are already well-established and known in the literature.
I address some of the claimed contributions and try to refute them as contributions next.

> We introduce CATS, a novel tree-based search algorithm that grows two trees, one forward from the initial state and one backward from the goal, coupled through a cross-tree guidance signal. Experiments show that CATS outperforms classic search methods on all tasks in Budget-BlocksWorld.

The bi-directional search algorithm proposed is known since the 60s [1]. Bi-directional search, in general, had a significant breakthrough in the last decade (winning awards at major conferences like AAAI, and IJCAI) [2,3,4,5,6], and has been a constant topic of study since the earlier paper by Pohl  [7,8,9]. Bi-directional search is even introduced in the earlier chapters of Russell & Norvig. It is known that for certain problems it outperforms forward search significantly.

One can say try to argue that the paper introduces new algorithms because it uses different names (e.g., it calls heuristics as rewards, for some reason). But this is just terminology and, speaking very directly, the mixing of RL/MCTS terms with simple state-space search algorithms is also a minor problem. The semantics of the state-space search are unchanged.

> However, most existing planning benchmarks like GSM8K (Cobbe et al., 2021), Game of 24, judge performance solely on plan correctness, largely ignoring real-world constraints such as time, cost, or resource limits. In practical applications, planning often operates under strict budget constraints, such as limited time steps, energy consumption, or financial cost, making it essential to
generate not only valid but also efficient plans (Xie et al., 2024; Zhang et al., 2025a). In this work, we introduce a cost-aware concept to the BlocksWorld benchmark that explicitly incorporates budget constraints into the planning process

The notion of action costs in planning is used and studied since decades, and even languages like PDDL support it. The same goes for the other constraints mentioned --- time, resource limit. Moreover, this passage does not make sense: for the bi-directional search, the size of the search space grows exponentially in the search depth as well. This can also lead to poor performance.

> To fill the gap in evaluating cost-sensitive planning, we proposed Budget-Blocksworld, a cost-augmented benchmark, along with
a complementary evaluation suite that measures plan success, cost efficiency, and search effectiveness.

The setting proposed, normally called cost-bounded planning, is widely studied in the planning literature, and the International Planning Competition has tracks aimed at this problem (e.g., satisficing, and cost-bound tracks) since the 2000s. (Note that the metrics used in the competitions are essentially equivalent to the ones in the paper.)

Needless to say, it is totally fine to use existing techniques. One could even say that the *combination* of these techniques is the main difference (which is not the case either, btw). But there are two problems here:

- when one knows the basic literature mentioned above, the results of this submission become trivial. LLMs are only used to compute the confidence reward (the search itself is done using PDDLGym), which is only a part of the heuristic function. Therefore this has only incremental value.
- this becomes a significant problem to me when the existence of previous work is never acknowledged in any manner, and these conclusions are presented as fresh new contributions.

Another major concern I have is regarding the interpretation of the results: the proposed method used a *domain-dependent heuristic* for Blocksworld. It is evident that this will speed-up the search and find shorter plans---this is the classic undergrad AI exercise to compare BFS/DFS with diverse costs and then best-first search with a specific heuristic, and then explain when best-first search will find a better plan. The ablation study should contain an experiment where the distance reward is removed all together, and the search occurs in a FIFO or LIFO ordering. Moreover, it was not clear to  me if the methods compared to made use of any domain-dependent information (apart from the LLM-extracted info).

I also have some minor problems that are worth pointing out:
- to have an unbiased result, you should run a fresh new domain that the LLMs did not see during training. It is expected that they were trained on traditional blocksworld, which is very similar, and so the LLMs are biased towards it, which might affect the results.
- section 2.3 seems wrong to me: "Recent advances have shown that LLMs and LLM-based frameworks can achieve remarkable performance on a variety of planning tasks" contradicts the work by Valmeekam et al. that you cite.
- the technique proposed only works in domain problems where the goal state is fully known. If the goal is just a condition (a partial state, a predicate, a set of states), then the method will not work or will require very expensive operations like subsumption. Similarly, if preconditions and effects of actions are not "conjunctive", one might have a problem. This limitation should be clearly stated.
- the paper should contain a standard symbolic baseline to put the results into context

Ref.:

[1] Ira Pohl.
Bi-directional and heuristic search in path problems. Stanford University, USA, 1969\
[2] Robert C. Holte, Ariel Felner, Guni Sharon, Nathan R. Sturtevant:
Bidirectional Search That Is Guaranteed to Meet in the Middle. AAAI 2016: 3411-3417\
[3] Robert C. Holte, Ariel Felner, Guni Sharon, Nathan R. Sturtevant, Jingwei Chen:
MM: A bidirectional search algorithm that is guaranteed to meet in the middle. Artif. Intell. 252: 232-266 (2017)\
[4] Joseph Kelly Barker, Richard E. Korf:
Limitations of Front-To-End Bidirectional Heuristic Search. AAAI 2015: 1086-1092\
[5] Vidal Alcázar, Patricia J. Riddle, Mike Barley:
A Unifying View on Individual Bounds and Heuristic Inaccuracies in Bidirectional Search. AAAI 2020: 2327-2334\
[6] Vidal Alcázar:
The Consistent Case in Bidirectional Search and a Bucket-to-Bucket Algorithm as a Middle Ground between Front-to-End and Front-to-Front. ICAPS 2021: 7-15\
[7] Dennis de Champeaux, Lenie Sint:
An Improved Bidirectional Heuristic Search Algorithm. J. ACM 24(2): 177-191 (1977)\
[8] Dennis de Champeaux:
Bidirectional Heuristic Search Again. J. ACM 30(1): 22-32 (1983)\
[9] T. Ikeda; Min-Yao Hsu; H. Imai; S. Nishimura; H. Shimoura; T. Hashimoto:
A fast algorithm for finding better routes by AI search techniques. Proc. Vehicle Navigation and Information Systems Conference, 291–296.

**Questions:**

Feel free to answer to my criticisms above, and correct any misconceptions that I might have.

---

### Official Review · Reviewer_18wr · 2025-10-31

**Soundness:** 1
**Presentation:** 3
**Contribution:** 2
**Rating:** 2
**Confidence:** 3

**Summary:**

This paper introduces Cost-Augmented Tree Search (CATS), a bidirectional search algorithm designed to address budget-constrained planning tasks with LLMs. The authors contribute a cost-augmented benchmark, "Budget-BlocksWorld," which extends the classic BlocksWorld environment with non-uniform action costs and budget constraints. CATS expands two search trees (forward from initial state, backward from goal) guided by confidence and distance rewards, aiming to find cost-efficient plans. Experiments show CATS outperforms raw LLMs (GPT-4.1, Claude-Opus-4.1) and single-tree baselines (ToT, RAP) across different budget constraints.

**Strengths:**

* The paper proposes to solve LLM planning research by incorporating cost constraints, which is a practical realistic problem.
* The framework is evaluated with multiple metrics, and the framework shows strong performance.
* The paper has clear and effective illustrations to illustrate concepts.
* The Budget Blocksworld extends the classical Blocksworld environment with action costs and budget constraints as an interesting variant.

**Weaknesses:**

* The Blocksworld environment has some special characteristics in that the inverse of actions exists in the action set. That is, stack <-> unstack, pick <->drop. This is special and not a general characteristic of planning problems. In a cooking scenario, you can cook an egg, but not uncook the egg. If the inverse of actions does not exist, how do the algorithm expand backwards? Even when the actions are theoretically reversable, the costs are often asymmetric. This is a major issue that limits the algorithm’s applicability to other tasks.
* The framework is only evaluated within the Blocksworld environment. This environment only has four actions, and the number of blocks is 6 for all problem instances. The generalizability to other planning domains with different characteristics remains unclear.
* The paper is missing ablation experiments that isolates the bi-directional tree search.

**Questions:**

* How can the algorithm be deployed if no inverse-actions exist in the action set?
* Can a single directional CATS be used to solve this problem?
* For the confidence and distance rewards, do they require task-specific efforts?

---

### Official Review · Reviewer_FWf4 · 2025-11-01

**Soundness:** 2
**Presentation:** 4
**Contribution:** 2
**Rating:** 4
**Confidence:** 4

**Summary:**

This paper addresses cost-awareness in LLM-based planning, where actions have costs and plans must follow budget constraints. The authors propose Cost-Augmented Tree Search (CATS), a novel bidirectional tree search method that expands both a forward-oriented tree starting from the initial state and a backward-oriented tree starting from the target state, which arguably "mitigate the exponential blowup in search" (line 128), while halting paths that accumulate cost beyond the allowed budget. CATS is tightly presented with a version of the BlocksWorld benchmark augmented with costs (Budget-BlocksWorld), which is the basis for experiments under three budget regimes: tight (must meet optimal cost), loose (can exceed optimal by a margin), and unlimited (no budget constraints). Experimental results span three dimensions (success rate, optimality, and efficiency), three pretrained models as chain-of-thought planners (Qwen3-8B, GPT-4.1 and Claude-Opus-4.1), more naive tree-search planning (tree-of-thoughts), and more intelligent tree-search planning (reasoning via planning), with CATS outperforming these baselines across conditions. Authors conclude with sensible ablations w.r.t. model size and reward weighting.

**Strengths:**

1. Very practically relevant topic.

2. The paper is very well written and easy to follow. The method design makes sense, and the bidirectional tree search is an interesting addition (although it limits applicability, see weakeness #1 below), whose implementation is clearly described.

3. The paper's experimental space (i.e., Section 5) is sensible and thoughtful, with the inclusion of both ToT and RAP with comparable rewards under the single tree search design (w = 1). Results analyzed over three dimensions (success rate, optimality, and efficiency) under three budget regimes are also largely clear (see weakeness #3 below) and positive.

**Weaknesses:**

1. Despite the practical relevance of the topic, the choice of artificially modifying BlocksWorld and restricting the paper to it makes CATS less compelling than what it could be. More specifically, authors could leverage a setting that already includes budget constraints such as TravelPlanner [1] or augmenting a setting where cost could play a more natural role, such as MBPP [2]. While cost-effectiveness would not appear artificial or arbitrary in these settings, the bidirectional design does reduce CATS' applicability by adding the requirement that the final state is known, which was not necessary for the study of cost-aware planning.

2. Compared to other planning papers (e.g., [2] or [3]), one single test setting is relatively limited. Moreover, the introduction of CATS in Section 4 as a general method should be decoupled from any individual setting (on line 209: "CATS adopts a tree search methodology to solve the Budget-BlocksWorld task").

3. Please add details on what "plus a small margin" means on line 194, for reproducibility.

[1] Xie, Jian, Kai Zhang, Jiangjie Chen, Tinghui Zhu, Renze Lou, Yuandong Tian, Yanghua Xiao, and Yu Su. "TravelPlanner: A Benchmark for Real-World Planning with Language Agents." In International Conference on Machine Learning, pp. 54590-54613. PMLR, 2024.

[2] Wang, Chaojie, Yanchen Deng, Zhiyi Lyu, Liang Zeng, Jujie He, Shuicheng Yan, and Bo An. "Q*: Improving multi-step reasoning for llms with deliberative planning." arXiv preprint arXiv:2406.14283 (2024).

[3] Hao, Shibo, Yi Gu, Haodi Ma, Joshua Hong, Zhen Wang, Daisy Wang, and Zhiting Hu. "Reasoning with Language Model is Planning with World Model." In Proceedings of the 2023 Conference on Empirical Methods in Natural Language Processing, pp. 8154-8173. 2023.

**Questions:**

1. On line 194: What does "plus a small margin" mean exactly?

2. How often do authors consider that the final state is known in planning tasks? Any insights on to what extent the paper can still be useful for this greater family of tasks?

---

### Meta-Review · Area_Chair_iAFe · 2026-01-06

**Summary:**

Reviewers pointed out that there are several crucial concerns with the paper:

1. There is limited novelty in this paper: Cost-aware planning is not a novel problem, and the tree search algorithm is a known technique.

2. The experiments are very limited, with only the BlocksWorld environment. The paper also lacks crucial ablation studies.

3. The backward expansion limites the applicability of the method.

**Reviewer Concerns:**

The reviewers did not engage in the rebuttal. So all concerns remain.

**Reviewer Scores:**

The reviewers did not engage in the rebuttal. So all scores should remain unchanged.

---

### Decision · Program_Chairs · 2026-01-26

Reject